# Two-Component Systems of *Streptomyces coelicolor*: An Intricate Network to Be Unraveled

**DOI:** 10.3390/ijms232315085

**Published:** 2022-12-01

**Authors:** Ricardo Sánchez de la Nieta, Ramón I. Santamaría, Margarita Díaz

**Affiliations:** Instituto de Biología Funcional y Genómica, Universidad de Salamanca, Consejo Superior de Investigaciones Científicas, 37007 Salamanca, Spain

**Keywords:** *Streptomyces*, two-component systems, signal transduction and regulation, molecular microbiology

## Abstract

Bacteria of the *Streptomyces* genus constitute an authentic biotech gold mine thanks to their ability to produce a myriad of compounds and enzymes of great interest at various clinical, agricultural, and industrial levels. Understanding the physiology of these organisms and revealing their regulatory mechanisms is essential for their manipulation and application. Two-component systems (TCSs) constitute the predominant signal transduction mechanism in prokaryotes, and can detect a multitude of external and internal stimuli and trigger the appropriate cellular responses for adapting to diverse environmental conditions. These global regulatory systems usually coordinate various biological processes for the maintenance of homeostasis and proper cell function. Here, we review the multiple TCSs described and characterized in *Streptomyces coelicolor*, one of the most studied and important model species within this bacterial group. TCSs are involved in all cellular processes; hence, unravelling the complex regulatory network they form is essential for their potential biotechnological application.

## 1. *Streptomyces*, the Biotech Gold Mine

Rapamycin, ivermectin, chloramphenicol, kanamycin, cycloheximide, and doxorubicin, among others, all key compounds for human medicine, are synthetized by bacteria of the genus *Streptomyces*. In addition to these compounds, a myriad of antibiotics, antifungals, antiparasitic agents, antivirals, immunosuppressors, antitumoral agents, and even multiple hydrolytic enzymes of great industrial value are produced by this group of microorganisms which act as an authentic biotech gold mine [1,2]. Moreover, owing to advances in sequencing genomes, it has been observed that these organisms contain a multitude of cryptic biosynthetic gene clusters (BCGs) that remain silent. As a consequence, their true potential as producers of compounds of biotechnological interest is much greater than initially estimated [3,4]. Based on these characteristics, much interest has been sparked in understanding the physiology of these bacteria, as well as unravelling the underlying regulatory mechanisms, that would allow for their manipulation and application.

At the taxonomic level, the genus *Streptomyces* forms part of the *Bacteria* superkingdom, the *Actinomycetota* phylum, the *Actinomycetia* class, the *Streptomycetales* order, and the *Streptomycetaceae* family [5,6]. These microorganisms are characterized as filamentous multicellular bacteria that are Gram-positive, facultatively aerobic, mesophilic, neutrophilic, and chemoorganotrophic, with an oxidative metabolism [7]. Concerning their life cycle, these bacteria develop a vegetative mycelium and an aerial mycelium that forms chains of immobile reproductive spores [7,8]. At the genomic level, these bacteria usually have a linear chromosome of 6–12 Mbp, with approximately 70% of GC content. The chromosomes are structured in a central region which is relatively conserved between different species and is where the essential genes are concentrated. The end of each chromosome is genetically unstable, with reorganizations, deletions, insertions, and duplications recurrently occurring [9,10].

The main habitat of Streptomycetes is soil, which is an extremely complex and dynamic environment exposed to a diverse array of biotic and abiotic factors. It is also here that Streptomycetes play an essential role in nutrient recycling as saprophytic organisms [7,8], owing to the genomic plasticity, metabolic diversity, and complex regulatory networks they have developed. However, they have also been found in other ecological niches, such as marine environments [11,12,13], and even form part of the microbiota of other organisms [14,15].

Additionally, both beneficial and harmful interactions with other organisms have been described. For example, certain insects use antibiotic-producing Streptomycetes to protect their resources [16,17], while some xylophagous beetles establish a commensalism relationship with some of these bacteria capable of degrading cellulosic substrates [18,19]. In the case of plants, some Streptomycetes which promote growth and act as biological control agents have been identified [20,21], whereas others are known to act as phytopathogens [22], all of which make these bacteria interesting from an agricultural perspective. Finally, although most of these bacteria are not harmful to humans, species such as *Streptomyces somaliensis* and *Streptomyces sudanensis* can cause mycetoma [23,24].

Within this group of microorganisms, *S. coelicolor* is one of the main model species, as it is one of the first to be morphologically and genetically characterized [25,26], and has been extensively studied at all levels during the past few decades. One of the main interests regarding this species is its applicability in understanding the regulatory mechanisms of antibiotic production. This species contains several synthesis clusters in its chromosome associated with actinorhodin (ACT), undecylprodigiosin (RED), and calcium-dependent antibiotic (CDA) production, of which the first two are colored, a characteristic that has been central to their experimental use.

## 2. Two-Component Systems: Essential Signal Transduction Systems for Bacteria

All organisms are exposed to a multitude of environmental and internal signals that must be detected, integrated, and processed if an organism is to appropriately react and adapt to its surrounding conditions. In response to these signals, numerous signal transduction mechanisms carry out this function, which in the case of prokaryotic organisms are predominantly the so-called two-component systems (TCSs) [27,28].

TCSs act as global regulatory systems able to detect external and internal stimuli, triggering cellular responses. TCSs are also involved in all the biological functions of the organism, from growth to intercellular communication, through different metabolic processes and responses to different types of stress. They generally act as pleiotropic regulators that coordinate various biological processes to maintain homeostasis and proper cell function [29].

As the name suggests, TCSs are usually composed of two components: a sensor element called histidine kinase (HK), responsible for the perception of certain signals, and an effector element called a response regulator (RR), responsible for generating a cellular response to these signals. In general, the signaling cascade (Figure 1) begins with the perception of a specific stimulus by the HK, which results in its activation and autophosphorylation in a histidine residue. Subsequently, the phosphoryl group is transferred to an aspartate residue of the RR, which triggers its activation and allows it to carry out its function and generate a cellular response [30].

HKs are usually embedded in the plasmatic membrane and form homodimers, although completely cytosolic HKs have also been described. The canonical architecture of an HK consists of a sensor domain, a transmembrane domain, a transmitter domain, a dimerization/phosphotransfer domain (DHp) containing the histidine residue that is phosphorylated, and an ATP-binding domain (CA) with kinase activity (Figure 1). The latter two domains constitute the catalytic core of an HK [27,28,31]. The variety of stimuli, both physical and chemical, that an HK can detect is immense, including temperature, light, oxygen concentration, different types of nutrients, and antibiotics, among other stimuli [32,33]. This is one of the reasons why the activation signals of most of the TCSs described are still unknown. Apart from this canonical architecture, there is another group of hybrid histidine kinases (hHKs) [30,32] that are characterized by the presence of additional phosphoacceptor receiver (REC), HPt, and/or DHp domains, which result in phosphotransfer events in the HK protein itself. The REC domain normally present in the RR has a phosphorylatable aspartate residue, while the HPt domain has a phosphorylatable histidine residue, although, unlike the DHp, they do not have a dimerization function. However, it is remarkable that the phosphotransfer process always takes place following the His→Asp→His→Asp scheme [30,32].

RRs are cytosolic proteins that contain two domains: a receptor domain (REC) and an effector domain (Figure 1) [28,31]. The REC domain is present in all RRs and contains the aspartate residue that is phosphorylated by the HK through phosphotransfer, a process that results in the activation of this protein. This domain is also responsible for controlling RR oligomerization processes that take place after its activation [28,31]. The phosphorylation site is defined by a group of highly conserved key residues, except in the case of atypical RRs (aRRs) that present alternative mechanisms of activation [34]. The effector domain triggers the cellular responses, normally through the modulation of gene expression since it is usually a DNA-binding domain. However, RNA-binding domains, protein-binding domains, and domains with enzymatic activity have also been identified (sometimes is even absent) [35,36].

Each HK is usually associated with a particular RR, forming a functional couple; normally, these HK-RR pairs are also together at the genomic level in the same operon [37]. The interaction between a HK and its associated RR is highly specific, which is key to proper cell function, since at certain times there may be a multitude of TCSs acting simultaneously. The specificity code depends on a set of amino acids mainly present in the DHp domain of the HK and in the REC domain of the RR, through which the interaction between the two proteins for the phosphotransfer process occurs [30,38]. This recognition at the molecular level establishes a kinetic preference between both proteins, which prevents non-specific interactions with other HKs and RRs [39,40]. Although crosstalk events between different TCSs have been described, these non-specific interactions do not appear to be relevant at the physiological level.

However, it is not uncommon to observe the presence of orphan HKs (oHKs) and orphan RRs (oRRs) in the bacterial genomes, for which there are no associated pairs at the genomic level; however, this does not mean that there are no associated pairs at the functional level [34]. An oHK and an oRR can be functionally associated with each other [41,42] or participate in the signaling cascade of a TCS as another component of the transduction route [43]. The prediction of the functional pairs of these oHKs and oRRs is still a serious obstacle, despite the efforts made using algorithms to predict protein–protein interactions.

TCSs are involved in the regulation and coordination of a multitude of biological processes. Since these signaling cascades must respond dynamically, there are different modulation mechanisms such as the control of the expression levels of HKs and RRs, the modification of the phosphorylation state of RRs, the cross-regulation of the TCS, nonlinear signaling cascades, and the regulation of the TCS through feedback from their targets [37]. All of these mechanisms modulate the amount of phosphorylated RRs, which is what defines the level of activation of a given TCS.

## 3. Two-Component Systems of *Streptomyces coelicolor*

The species *S. coelicolor* presents a large number of TCSs, specifically 110 HKs and 87 RRs, based on the P2CS (http://www.p2cs.org/; accessed on 27 May 2022) and P2RP (http://www.p2rp.org/; accessed on 3 June 2022) databases. The HKs have been classified based on their architecture, being either classic or hybrid (Figure 2), while RRs have been classified based on the type of effector domain they contain (Figure 2). These can include DNA-binding domains (NarL, OmpR, IclR, and LytTR), an RNA-binding domain (AmiR), enzymatic activity (RssB, TrxB, and ATPase), or the absence of an effector domain (REC only).

These proteins are organized into 66 HK-RR pairs and 2 triads (one of which two HKs and one RR and the other one with one HK and two RRs). The remaining proteins remain as orphan systems: 41 oHKs and 18 oRRs. The complete list is shown in Table 1.

The different TCSs described in the literature are detailed below. They are listed in alphabetical order and there is a focus on their biological role and/or molecular mechanisms of action.

### 3.1. AbrA1/A2 (SCO1744/45)

This system has been described as a negative regulator of antibiotic production (ACT, RED, and CDA) and morphological differentiation [44]. The AbrA1/A2 TCS can self-regulate its expression since it acts as a positive regulator of its operon [45]. In addition, it has been observed that high levels of the AbrA2 RR in its phosphorylated state are toxic to the cell and that it is the AbrA1 HK that is responsible for maintaining an adequate balance of the level of phosphorylation of the RR through its phosphotransfer/phosphatase activities [45].

At the genomic level, this TCS forms an operon with a potential ABC transport system encoded by the *SCO1742/43* genes, which seems to be involved in its mechanism of action, although its role has not been determined.

At the biotechnological level, it is important to note that the strain lacking this TCS can act as a host strain with an improved capacity to produce heterologous compounds of interest, such as oviedomycin, which present antitumor and antibiotic activities [45].

It has been proposed that the activation signal of AbrA1 is iron since the self-induction capacity of this TCS depends on the level of this element [45].

### 3.2. AbrB1/B2 (SCO2165/66)

This system has been described as a negative regulator of antibiotic production (ACT and RED) and a positive regulator of vancomycin resistance [46].

This TCS is a homolog of VraR/S (*Staphylococcus aureus*) and LiaR/S (*Enterococcus faecium*), which act as sensors of cell wall damage and are involved in the cell envelope stress response, playing a key role in clinically relevant antibiotic-resistant bacteria [46].

### 3.3. AbrC1/C2/C3 (SCO4598/97/96)

This atypical TCS consists of two HKs and one RR and has been described as a positive regulator of antibiotic production (ACT, RED, and CDA) and morphological differentiation [44,47]. It has been determined that the AbrC3 RR directly controls the specific regulator of the ACT synthesis pathway *actII-ORF4*, as well as other genes involved in different processes of primary metabolism, secondary metabolism, and morphological development; in addition, a potential binding sequence has been identified [47]. This system presents a cross-regulation with the AfsQ2/Q1 TCS [48].

Both HKs (AbrC1 and AbrC2) show similar expression levels; however, the elimination of AbrC2 does not present clear phenotypic effects, unlike the elimination of AbrC1 which produces an increase in the production of antibiotics. These observations can be attributed to differences in the kinase/phosphatase functions of these HKs, although it has been seen that both can phosphorylate the AbrC3 RR [49]. In addition, it has been determined that high levels of AbrC3 in its phosphorylated state are toxic to the cells [49].

The fusion of the HKs AbrC1 and AbrC2, due to a deletion that covers part of the sequence of both genes (although maintaining the reading frame), gives rise to a chimeric HK called LinR. This mutation results in high resistance to lincomycin (an antibiotic that binds to ribosomes and inhibits protein synthesis), increases the production of the antibiotic ACT, and enhances morphological differentiation [50]. The heterologous expression of LinR in *Escherichia coli* confers resistance to lincomycin in this organism [50].

### 3.4. AbsA1/A2 (SCO3225/26)

This TCS was one of the first regulators of antibiotic production described in [51]. This system has been established as a negative regulator of the production of the antibiotics ACT, RED, and CDA [52,53,54,55,56], through the direct regulation of the specific regulators of the synthesis pathways *actII-ORF4*, *redZ,* and *cdaR* [55,57]. In the case of the regulation of CDA synthesis, it has been observed that AbsA2 is also able to directly regulate several genes of the *cda* cluster [55]. Furthermore, it has been found that this regulation depends on the phosphorylation state of the AbsA2 RR, which is controlled by the balance of kinase/phosphatase activities of the AbsA1 HK [58].

The AbsA1/A2 TCS can self-regulate its expression since it acts as a positive regulator of its operon [53,55]. However, it is important to note that different promoters have been identified in this operon, one of which involves the transcription of the complete operon. Additionally, there is another internal promoter that only regulates the expression of the AbsA2 RR, which explains the high levels of RR expression detected as compared to the HK [59].

The transmembrane topology of AbsA1 HK has also been studied due to its atypical structure. It has been determined that this protein has five transmembrane domains, four of which are located at the N-terminus and the rest at the C-terminus. In fact, the extracellular portion of the protein (which probably includes the sensor domain) is located at the C-terminal end [60].

The biotechnological potential of this TCS has been demonstrated through the induction of antimicrobial activities in different Streptomycetes through the heterologous expression of the AbsA1 HK of *S. coelicolor* [61]. The design of this induction strategy is interesting in that the heterologous expression of AbsA1 can counteract the effect of its orthologous, leading to the dephosphorylation of the homologues of the AbsA2 RR, which should act normally as negative regulators of the production of antibiotics. This in turn can improve the expression of some genes of secondary metabolism [61].

### 3.5. AfsQ2/Q1 (SCO4906/07)

This system is involved in the regulation of primary metabolism, secondary metabolism, and morphological development [48,62,63,64].

The effect of this TCS depends largely on the medium and growing conditions. It was initially identified for its involvement in the regulation of the production of ACT and RED antibiotics in *Streptomyces lividans* [63]. Later studies have identified this system as a positive regulator of the synthesis of the antibiotics ACT, RED, and CDA (through the specific pathway regulators *actII-ORF4*, *redD*, *redZ,* and *cdaR*), and a negative regulator of morphological development, in the presence of glutamate as the only source of nitrogen [64]. AfsQ2/Q1 has also been shown to be a positive regulator of other compounds of secondary metabolism such as coelimycin P2, a positive regulator of genes involved in phosphate and carbon metabolism, and a negative regulator of genes involved in nitrogen metabolism (mainly those involved in nitrogen assimilation) [48,62].

The regulation of nitrogen metabolism seems to be key in the AfsQ2/Q1 signaling pathway, as it is involved in maintaining homeostasis in nutrient utilization under conditions of high glutamate concentrations, establishing cross-regulation with oRR GlnR (a key regulator of nitrogen metabolism to be later discussed) [48].

The AfsQ2/Q1 signaling pathway also involves the sigma factor SigQ, encoded by the adjacent gene *SCO4908*, which is regulated by the AfsQ1 RR [64].

A final aspect to consider is that this TCS forms an operon with an additional gene, *SCO4905*, which encodes the potential lipoprotein AfsQ3 that could be involved in the signaling cascade through interaction with the HK in a similar way to other TCSs [32].

### 3.6. Aor1 (SCO2281)

This oRR acts as a positive regulator of antibiotic production (ACT, RED, and CDA) and the morphological differentiation process [65]. Transcriptomic studies have revealed that Aor1 is also involved in other cellular processes such as the protein secretion pathway via sigma factor SigU and in the osmotic stress response via the sigma factor SigB pathway [65]. The high number of altered genes in these studies suggests that Aor1 could be a master regulator within the physiology of this organism, although its signaling cascade has not yet been elucidated.

### 3.7. BldM (SCO4768)

This oRR was initially called WhiK [66,67]. This protein acts as a central regulator of the morphological development and differentiation of aerial mycelium [66,67,68,69]. BldM is regulated by the extracytoplasmic function (ECF) sigma factor BldN [68].

Although BldM presents a typical phosphorylation site, it has been observed that the phosphorylation of this RR is not required for its function, so it may present alternative activation mechanisms such as aRR [66].

BldM shows different mechanisms of action. On the one hand, BldM can form homodimers that regulate a certain set of genes independently of the *whi* regulation pathway (involved in the sporulation process). On the other hand, BldM can form heterodimers with the oRR WhiI (which will be discussed later) and regulate a different set of genes. The second mechanism takes place after the first, which agrees with the stages of the life cycle (sporulation takes place once the aerial mycelium is developed) and allows the integration of key regulatory routes in the process of development and morphological differentiation [70].

### 3.8. ChiS/R (SCO5378/77)

This system is involved in the regulation of the chitinase enzyme ChiC in response to the presence of chitin [71,72]. However, the deletion of ChiR does not have an overall effect on chitinase activity, which may be due to the presence of multiple chitinases in this organism [72].

### 3.9. CseC/B (SCO3359/58)

This system is essential for maintaining cell wall integrity through the ECF sigma factor SigE [73,74]. The deletion of *sigE* or *cseB* produces a similar effect, consisting of increased sensitivity of the cell wall to lytic enzymes (due to an altered peptidoglycan profile), increased production of the antibiotic ACT, and a sporulation deficit, dependent on magnesium levels in the medium [74]. At the genomic level, although SigE (*SCO3356*) presents a monocistronic transcription, it can also be transcribed as part of the *cseA* (*SCO3357*), *cseB*, and *cseC* operon, through a positive self-regulation process mediated by the CseB RR [74].

To determine the activation signals of CseC/B, the construction of a bioassay system based on the regulation of this operon was carried out [73,75], which identified inducer compounds of this TCS that affect the integrity of the cell wall such as glycopeptide antibiotics, β-lactam antibiotics, and lysozymes. Based on these results, it has been proposed that the CseC/B TCS is activated by the accumulation of the intermediate products of peptidoglycan synthesis/degradation [73]. The perception of these signals by this system also involves the lipoprotein CseA, which is anchored to the plasma membrane and oriented towards the extracytoplasmic space, and probably modulates the activity of the CseC HK through interaction with its sensor domain, although the mechanism of action is unclear [76].

### 3.10. CssS/R (SCO4155/56)

It is important to note that this system has been described in *S. lividans*, a species extremely close to *S. coelicolor*, and has been included in this review for this reason.

The CssS/R TCS is involved in the secretion stress response that is caused by the accumulation of misfolded proteins on the outside of the plasmatic membrane which can interfere with the secretion machinery. It has been observed that the CssS/R system is induced by the overproduction of α-amylase, which causes secretion stress; in response to this, CssR regulates the proteases of the HtrA family for the elimination of misfolded proteins [77].

### 3.11. CutS/R (SCO5863/62)

This system acts as a negative regulator of the production of the antibiotic ACT, although it seems to be indirect as it does not bind to the specific regulator of the *actII-ORF4* pathway [78].

### 3.12. Cvn (Conservons)

When the genome of *S. coelicolor* was described [25], 13 conservons were identified and called Cvn1 to 13 (the name derives from conserved operons). All conservons have a similar organization, consisting of four or five genes with specific functions: CvnA is an oHK, CvnB is a protein of the *roadblock/LC7* family, CvnC is a protein of unknown function, CvnD is a GTP/GDP-binding protein, and CvnE is a cytochrome P450. These proteins can form complexes similar to eukaryotic G-protein-coupled receptors (GPCRs).

The oHK CvnA1 (*SCO5544*) acts as a positive regulator of antibiotic production (ACT and RED) and the development of aerial mycelium through sigma factor SigU [79].

The oHK CvnA9 (*SCO1630*) acts as a negative regulator of the production of the antibiotic ACT and the development of aerial mycelium through the *bld* pathway [80]. It has been determined that this oHK interacts with CvnB9 and CvnC9, which in turn interact with CvnD9 [80].

The oHK CvnA10 (*SCO7422*) exhibits similar behavior to CvnA9 and acts as a negative regulator of the production of the antibiotic ACT and the development of aerial mycelium [80].

### 3.13. DraK/R (SCO3062/63)

This system has been described as a regulator of antibiotic production, morphological differentiation, and primary metabolism [81,82]. Regarding the production of antibiotics, DraK/R acts as a positive regulator of ACT synthesis (controls directly the *actII-ORF4* regulator) and a negative regulator of RED synthesis under conditions of high levels of different nitrogen sources (such as glutamine, glutamate, or glycine among others) [81]. The DraK/R TCS can self-regulate its expression since it acts as a negative regulator of its own operon [81]. The regulation of antibiotic production exerted by DraK/R is conserved in other Streptomycetes, such as *Streptomyces avermitilis*, where it acts as a negative regulator of avermectin synthesis and a positive regulator of oligomycin synthesis [81].

The study of the sensor domain of the DraK HK, to try to determine the detection and signaling mechanism of this regulation system, is noteworthy [83,84,85]. The study of the biochemical and biophysical properties of this domain has revealed that it undergoes pH-dependent conformational changes that could be key in the signal transduction process of this TCS [83]. In addition, the structure of this domain (PDB ID: 2MJ6) has been obtained and led to the identification of the key residues that drive conformational changes in response to variations in pH [84]. Based on these studies, it has been proposed that the DraK/R TCS could play a key role in regulating the pH of the medium which *S. coelicolor* grows in [84].

### 3.14. EcrA1/A2 (SCO2518/17)

This system has been described as a positive regulator of the antibiotic RED [86].

### 3.15. EcrE1/E2 (SCO6421/22)

This system acts as a positive regulator of the production of the antibiotic RED, through the specific pathway regulators *redD* and *redZ* [87].

### 3.16. GlnR (SCO4159)

This oRR acts as a central regulator of nitrogen metabolism and is one of the most extensively studied RRs of *S. coelicolor* [88,89,90,91,92,93,94,95,96,97,98]. Besides its role in this key cellular process, GlnR is involved in many other directly or indirectly associated processes, such as carbon metabolism [93,99], antibiotic synthesis [93,100], or osmotic stress response [101].

GlnR controls the expression of many key genes at different stages of nitrogen metabolism such as glutamine synthesis [90,96,97], ammonium assimilation [91,94], nitrate/nitrite assimilation [89], nitrite reduction [94], nitrate reduction [95], urea degradation [94], and amino acid biosynthesis [93]. All of these studies have made it possible to identify the target DNA sequences to which GlnR binds.

The oRR GlnR interacts in cross-regulation with other TCSs, such as MtrB/A [102], AfsQ2/Q1 [48], and PhoR/P [103,104,105,106], of which GLnR competes in the regulation of the main genes involved in nitrogen metabolism. This in turn allows GLnR to participate in other key processes such as phosphate metabolism.

Several reviews on nitrogen metabolism in *S. coelicolor* and its relationship with other primary metabolic processes have been published [107,108,109].

Regarding the mechanism of action at the molecular level of GlnR, certain alterations have been observed at the phosphorylation site as compared to the canonical structure of its RR. Furthermore, it has been determined that GlnR can form homodimers in the absence of the phosphorylation of the aspartate residue, a structure that is stabilized through electrostatic interactions established by this aspartate with other residues [92]. GlnR is modified post-translationally by the phosphorylation of the serine and threonine residues, as well as the acetylation of lysine residues [88]. These modifications vary depending on the levels of nitrogen and other nutrients in the medium (the higher the nitrogen levels in the medium, the greater the degree of phosphorylation in Ser/Thr of GlnR) and modulate the binding affinity of this RR to DNA, which affects the transcriptional response generated by it [88].

### 3.17. GluK/R (SCO5779/78)

This system is involved in the detection and uptake of glutamate, since it regulates the *gluABCD* operon (*SCO5774-77*) that is adjacent at the genomic level and encodes the glutamate uptake system, under conditions of high levels of this compound [110].

In addition, GluK/R acts as a negative regulator of ACT synthesis, and a positive regulator of RED synthesis. However, the role of this TCS in regulating antibiotic production is independent of its role in glutamate uptake [110].

This TCS is one of the few cases in which the HK activation signal has been determined. A biolayer interferometry (BLI) assay has shown that it is the glutamate molecule that interacts with the GluK HK and acts as its activation signal [110]. Structurally similar compounds, such as glutamine, are not recognized by this HK.

### 3.18. MacS/R (SCO2121/20)

This system has been described as a positive regulator of antibiotic production (ACT, RED, and CDA) and a negative regulator of morphological differentiation [111,112]. Multiple direct targets of MacR have been identified, i.e., membrane proteins and lipoproteins which seem to act as morphogenetic factors that participate in the development of aerial mycelium. In addition, the binding consensus sequence of this RR has been established and validated [112].

### 3.19. MtrB/A (SCO3012/13)

Much interest is focused on this TCS, owing to the orthologous system present in *Mycobacterium tuberculosis*, which seems to be involved in the regulation of the osmotic stress response, the homeostasis of cell envelopes, and the progression of the cell cycle (coordinating DNA replication with cell division) [113].

In *S. coelicolor*, it has been described that the MtrB/A TCS is essential in the development of aerial mycelium, as it acts as a positive regulator of many key genes in the differentiation process of the *chp*, *rdl*, *ram*, *bld,* and *whi* families [114].

In addition, this system acts as a key regulator in the production of antibiotics, as well as a negative regulator of the synthesis of ACT and RED and as a positive regulator of the synthesis of CDA. In all cases, MtrB/A exerts direct regulation on the specific regulators of the synthesis pathways of these antibiotics such as *actII-ORF4*, *redZ*, and *cdaR* [115,116]. The implication of MtrB/A in the regulation of antibiotic production is well preserved in other Streptomycetes, as it is involved in the control of biosynthetic clusters of chloramphenicol and jadomycin in *Streptomyces venezuelae*, avermectin, and oligomycin in *S. avermitilis*, and validamycin in *Streptomyces hygroscopicus* [116].

The MtrB/A TCS has also been associated with nitrogen metabolism [102] and phosphate metabolism [117] in *S. coelicolor*. It has been described that MtrA represses nitrogen assimilation genes in nitrogen-rich media, and GlnR (key regulator in nitrogen metabolism) under limiting conditions. In addition, MtrB/A competes with this latter oRR in binding to its target sequences [102]. Thus, MtrA and GlnR compete in the control of genes involved in nitrogen metabolism, with priority placed on regulation by GlnR under nitrogen-limiting conditions, and a predominating regulation by MtrA in nutrient-rich conditions [102]. MtrA also regulates the key genes of phosphate metabolism, including the PhoP/R TCS, under different conditions (both in limiting and phosphate-rich situations); this regulation is similar to that of PhoP [117]. In addition, nitrogen metabolism is directly related to phosphate metabolism through these regulation systems, as it has been observed that the MtrB/A and PhoP/R systems regulate different nitrogen metabolism genes, including GlnR, under different phosphate availability conditions [117]. In summary, a clear phenomenon of the cross-regulation between the MtrB/A, PhoP/R, and GlnR systems can be observed concerning nitrogen and phosphate metabolism.

### 3.20. OhkA (SCO1596) and OrrA (SCO3008)

The oHK OhkA has been described as a negative regulator of antibiotic production (ACT, RED, and CDA) and a positive regulator of aerial mycelium formation and the sporulation process [118,119].

Recent studies have identified the RR related to OhkA, which forms part of its signal transduction pathway, and have determined that it is the oRR encoded by the *SCO3008* gene called OrrA [42]. The identification of OrrA as an OhkA-associated RR has been conducted through phenotypic, transcriptomic, and double-hybrid experiments. In fact, the removal of OrrA produces an effect similar to the deletion of OhkA [42]. Moreover, previous studies have linked oRR OrrA to the regulation of antibiotic synthesis [120].

This signal transduction system comprising OhkA and OrrA also involves the pleiotropic regulator WblA (*SCO3579*). It has been determined that the expression of WblA is directly regulated by OrrA [121]. The role described for WblA is similar to that of OrrA, known to be a negative regulator of antibiotic production and a positive regulator of morphological development process [122,123,124].

### 3.21. OsaA/B (SCO5748/49)

This system is involved in the osmotic stress response since the OsaB RR is required for the osmoadaptation needed during the differentiation process. In addition, it is involved in the production of antibiotics acting as a negative regulator of the synthesis of ACT and RED [125].

The *osaB* gene has been determined to have its own promoter, independent of *osaA* [125].

The OsaC regulator (*SCO5747*), which acts as an anti-sigma factor, is related to the regulation of the OsaB RR. OsaC is induced after exposure to osmotic stress by sigma factor SigB (a key factor in the response to osmotic stress and oxidative stress, as well as in the processes of differentiation and production of antibiotics) and is required to restore the expression levels of OsaB and SigB once the response to osmotic stress has taken place [126].

A noteworthy feature of the OsaA and OsaB proteins is their atypical architecture. OsaA has been classified as a hybrid HK because it has a REC domain (with a phosphorylable aspartate residue). Another interesting structural characteristic is the presence of more than ten HAMP domains (this protein is very large, with a length of 1829 aa). In the case of the OsaB RR, its effector domain does not belong to any of the canonical families; this domain has a supercoiled helix structure (coiled coil) that is believed to interact with similar motifs of other proteins.

### 3.22. OsdK/R—DevS/R (SCO0203/04)

The name of the TCS encoded by *SCO0203/04* presents certain discrepancies in the literature; some authors have named this system OsdK/R [127,128], while others refer to it as DevS/R [129,130].

The initial interest in this system arose because it is an ortholog of the DevS/R TCS (also called DosS/R) of the human pathogen *M. tuberculosis*, involved in the activation of the dormancy state after hypoxia conditions [131]. The state of dormancy consists of a cessation of growth, which is hugely important in pathogenic bacteria. This condition allows these bacteria to resist the defense system of the host, as well as antibiotic treatment, playing a central role in latent infections. Moreover, it has been described that the OsdK/R TCS of *S. coelicolor* controls a regulon associated with a state of dormancy as it includes a multitude of genes related to stress response and development [128].

In addition to this role, OsdK/R participates in the cycle of nitric oxide of *S. coelicolor* (gas that acts as an important signaling molecule in the control of bacterial metabolism, although it is toxic at high levels), which allows the interconversion of nitrates, nitrites, and nitric oxide through a dioxygenase (Fhb) and a nitrate reductase (Nar), the latter of which is regulated by this TCS. This cycle regulates endogenous nitric oxide levels, ACT antibiotic production, and morphological differentiation [130]. In this situation, OsdR acts as a positive regulator of ACT production, through the direct regulation of the specific regulator pathway *actII-ORF4* [129].

The role of OsdK/R In both processes is not incompatible, especially considering that oxygen is one of the major determinants of nitric oxide metabolism. In fact, it has been described that the synthesis of NarG2 nitrate reductase is induced by hypoxia conditions through OsdK/R in the mycelium of *S. coelicolor* [127]. It has been proposed that the stimuli perceived by OsdK could be precisely the levels of oxygen and nitric oxide, which would be detected through a heme group, similar to its DevS ortholog in *M. tuberculosis*.

Apart from this canonical signaling cascade, it has been determined that the OsdK HK is also able to phosphorylate the oRR *SCO3818* (in addition to its associated RR, OsdR, and its autophosphorylation, as evidenced by in vitro phosphorylation assays). In addition, at the phenotypic level, it has been observed that OsdK and oRR *SCO3818* could be involved in the downregulation of ACT antibiotic production under certain conditions [43].

### 3.23. PdtaS (SCO5239) and PdtaR (SCO2013)

This signal transduction pathway consists of the oHK PdtaS and the oRR PdtaR. PdtaS/R acts as a negative regulator of antibiotic production (ACT and RED) and a positive regulator of the process of differentiation [41,132].

The identification of PdtaR as the RR associated with PdtaS was initially established through bioinformatics but was later confirmed using in vitro phosphotransfer and phenotypic assays [41].

Interesting features of oRR PdtaR include its architecture, whose effector domain belongs to the AmiR family, and its ability to regulate RNA (transcriptional antiterminator).

### 3.24. PhoR/P (SCO4229/30)

This system acts as a central regulator of phosphate metabolism and is one of the most extensively studied TCSs in *S. coelicolor* [28,103,104,105,133,134,135,136,137,138,139,140,141,142,143,144,145,146,147]. Phosphate metabolism has a huge impact on many other cellular processes. Consequently, PhoR/P is also directly and indirectly involved in these processes which include the production of antibiotics [133,139,140,142,143,144], nitrogen metabolism [103,105,106,133,139], carbon metabolism [133,139], and morphological differentiation [133,139,140], among others.

The PhoR/P TCS responds to the phosphate-limiting conditions and controls the expression of many key genes at different stages of phosphate metabolism, such as phosphatases involved in phosphate recycling processes [105,134] or phosphate uptake systems [139,141,143,146]. In addition, this TCS is self-regulating, since it acts as an inducer of its operon [135,146], and presents a feedback control mechanism through PhoU (*SCO4228*), which acts as a negative regulator of the PHO regulon and is activated by PhoR/P [137].

All of these studies have facilitated target DNA sequences to which PhoR binds to be identified, which are called PHO-Boxes. There are different types of these binding sequences, depending on the number of repetitions of the consensus motif (Dru) and its organization, which affect the type of regulation exercised by PhoP and its degree of affinity [145,148].

The PhoR/P TCS presents a cross-regulation with other TCSs such as GlnR [103,104,105,106].

It has recently been proposed that the difference in PhoR/P levels between closely related *S. coelicolor* and *S. lividans* (this TCS is less abundant in *S. coelicolor*) could explain the metabolic differences between these two species [136,138].

In summary, the PhoR/P TCS acts, under limited phosphate conditions (essential resource for proper cell functioning), as a central regulator of cellular processes, blocking much of the primary metabolism, secondary metabolism, and development and differentiation pathways until the phosphate levels recover, and so the growth and proper functioning of the organism can resume.

Several reviews on phosphate metabolism in *S. coelicolor* and its relationship with other cellular processes have been carried out, as well as the PhoR/P transduction pathway [108,148,149,150,151].

### 3.25. RagK/R (SCO4073/72)

This system participates in the developmental processes of aerial mycelium and sporulation [152].

The process of morphological differentiation in *S. coelicolor* is complex and is largely controlled by the oRR RamR (which will later be discussed) and the morphogenic peptide SapB, which promotes the formation of the aerial mycelium by breaking the surface tension of the medium. One of the clusters regulated by RamR is *rag*, which contains the RagK/R TCS. The deletion of the *rag* cluster results in alterations in the morphology of the aerial mycelium, as well as in the processes of differentiation, septation, and sporogenesis. It has been proposed that RagK/R (together with the rest of the genes comprising the *rag* cluster) constitute a developmental pathway related to aerial mycelium and sporulation that is independent of SapB and which allows the integration of various morphogenic changes [152].

### 3.26. RamR (SCO6685)

This oRR acts as a central regulator of the process of morphogenesis during the development and differentiation of this organism [153,154]. RamR is kept within the *bld* signaling pathway of the development process and acts as a regulator of SapB, a morphogenetic peptide that is involved as a surfactant in the development of aerial mycelium [153,154].

### 3.27. RapA2/A1 (SCO5404/03)

This system acts as a positive regulator of the antibiotic ACT and coelimycin through pathway-specific regulators (*actII-ORF4* and *kasO*) [155].

### 3.28. RedZ (SCO5881)

RedZ is an atypical oRR that presents a series of functional and structural peculiarities. It is part of the biosynthetic *red* cluster and acts together with RedD as a pathway-specific regulator. RedZ acts as a positive regulator of RedD, which in turn activates the biosynthetic genes of the cluster and therefore the production of the antibiotic RED. In addition, RedZ presents negative self-regulation [156]. The dependence of the growth process presented by the production of the antibiotic RED seems to be given by the regulation of RedZ, which contains a rare codon of leucine UUA, and requires *bldA* (which encodes the only tRNA of this codon) for translation [156,157].

On a structural level, RedZ is an atypical RR and lacks the phosphorylation site of the REC domain [157]. The alternative control mechanism presented by RedZ seems to be given by its interaction with the molecule of the antibiotic RED, i.e., the final product of its signaling pathway [158].

### 3.29. SatK/R (SCO3390/89)

This TCS acts as a regulator of the sporulation process and morphological development [159].

In *S. coelicolor*, as in other organisms, chromosomal topology plays a critical role in gene regulation; this is determined by chromosomal supercoiling (controlled primarily by TopA topoisomerase) and nucleoid-associated proteins (NAPs). In this organism, TopA is essential, and the decrease in its levels produces an increase in the supercoiling of DNA and the alteration of the expression of many genes (among them, the supercoiling hypersensitive clusters (SHCs) stand out), giving rise to a significant reduction in growth and blocked sporulation. The SatKR TCS was discovered because mutations in this system were able to suppress the blocking of sporulation associated with the high supercoiling of DNA by reduced levels of TopA [159].

When SatR levels are low, e.g., due to the absence of SatK, this RR inhibits supercooling-dependent SHC activation. However, when SatR levels are high through SatK activation, SHC transcription is induced independently of supercooling, resulting in blocked sporulation and growth [159].

### 3.30. SenS/R (SCO4275/76)

This system is involved in the oxidative stress response [160].

The SenS/R TCS acts in conjunction with HbpS (*SCO4274*), a secreted protein. The signaling pathway formed by these three components appears to be modulated by post-translational modifications due to redox stress. The genes regulated by this system have been identified and form the non-enzymatic response against the effects of oxidative stress, since they intervene in the synthesis of compounds that act as redox buffers and scavengers of reactive oxygen species, although enzymes with catalase or superoxide dismutase activity are not involved [160].

### 3.31. SitK/R (SCO4667/68)

This system was described in parallel with the SatKR TCS (previously discussed), as it is part of the supercoiling hypersensitive cluster (SHC) and is involved in the inhibition of sporulation [159]. It has been observed that the removal of SitKR can restore blocked sporulation due to DNA supercoiling conditions [159].

### 3.32. VanS/R (SCO3589/90)

This system controls resistance to the antibiotic vancomycin [161] and is one of the most extensively studied TCS in *S. coelicolor*.

Vancomycin is a glycopeptide antibiotic that inhibits cell wall synthesis by binding to peptidoglycan precursors (binds to the D-Ala-D-Ala end of Lipid II), blocking the formation of cross-links necessary to generate a functional cell wall. Vancomycin only affects Gram-positive bacteria (it cannot cross the outer membrane present in Gram-negative bacteria), and its clinical importance is key, as it is the only effective treatment against *S. aureus* MRSA. The vancomycin resistance cluster (*van*) is present in multiple human pathogens (*S. aureus*, *E. faecium*, etc.) and in several actinomycetes (glycopeptide antibiotic producers such as *Amycolatopsis orientalis*, *Streptomyces toyocaensis*, etc., and non-producers such as *S. coelicolor*) [162].

In *S. coelicolor*, the *van* cluster consists of seven genes, vanSRJKHAX (*SCO3589/90; SCO3592-96*), and is controlled by the VanSR TCS which forms part of the cluster [161]. The VanS activation signal is the vancomycin molecule itself, and not intermediates of peptidoglycan synthesis, as in other organisms. In addition, it has been determined that the binding sites of vancomycin to VanS and Lipid II are distinct, and the antibiotic molecules recognized by the VanS HK must be bound to this peptidoglycan precursor for the activation of the signaling cascade [163,164,165,166,167]. VanS autophosphorylates His150 and catalyzes the phosphorylation/dephosphorylation of VanR in Asp51. In the absence of vancomycin, VanS acts as VanR phosphatase (which can be phosphorylated via acetyl-phosphate under these conditions). In the presence of vancomycin, VanS goes on to act as a kinase, allowing the activation of VanR, which in turn activates the entire vancomycin resistance cluster [163]. The interaction between VanS and VanR seems to require additional residues of the RR beyond the REC domain [167].

Vancomycin resistance is achieved by modifying cell wall synthesis, replacing the D-Ala-D-Ala end of lipid II with D-Ala-D-Lac, which is not recognized by this antibiotic. This substitution is carried out by VanHAX. VanH is a lactate dehydrogenase enzyme that transforms pyruvate into D-lactate, VanX is a dipeptidase that breaks down D-Ala-D-Ala peptides, and VanA is a ligase that binds D-Ala and D-Lac. In *S. coelicolor*, the cluster also contains VanJ, whose function is unknown, and VanK, which is an enzyme responsible for binding a glycine branch to the modified precursor Lipid II (in the unmodified precursor, the reaction is catalyzed by FemX) [161,168].

Recently, it has been possible to obtain the structure of the VanR RR by X-ray diffraction, both in its inactive form (PDB ID: 7LZ9) and in its active form (PDB ID: 7LZA). The structural analysis of this protein has made it possible to identify the conformational changes produced by its activation via phosphorylation, which enables the dimerization of the REC domain. Although the inactive conformation seems able to bind to DNA, it has been proposed that the dimer of this RR presents a greater affinity for binding [169].

This system presents a cross-regulation with the AbrB1/B2 TCS [46] and CseC/B [161].

Finally, it should be noted that resistance to vancomycin in *S. coelicolor* is dependent on phosphate levels in the medium; this process is independent of the PhoR/P TCS and seems to be related to the phosphate content of the polymers present in the cell wall [170,171,172,173].

### 3.33. WhiI (SCO6029)

This oRR is one of the central regulators of the sporulation process and seems to play a key role in the process of septation of the aerial mycelium required for sporulation, as well as spore maturation [174,175,176,177]. WhiI is regulated by WhiG, the specific sigma factor of sporulation; in addition, it has been determined that this oRR is negatively self-regulated [178].

On a structural level, WhiI is an aRR since it lacks the phosphorylation site of the REC domain [176,178].

As previously mentioned, WhiI can form functional heterodimers with the oRR BldM, which integrates the development processes of aerial mycelium and sporulation [70].

### 3.34. SCO5282/83

This system is involved in the regulation of different processes related to extracellular metabolism (protein secretion, carbon metabolism, and the metabolism of cell envelopes, among others) [179].

The initial interest in this TCS is given because certain mutations in the HK give rise to significant morphological changes in liquid medium, which could be favorable for its growth in industrial fermenters [179].

### 3.35. SCO5351

This oRR acts as a positive regulator of antibiotic production (ACT and CDA) and the processes of aerial mycelium development and sporulation [180]. In addition, some key residues in the oligomerization process of this oRR, whose mutation blocks its action, have been identified [180].

### 3.36. SCO5784/85

This system acts as a positive regulator of the production of ACT and RED antibiotics, the sporulation process, and secreted proteins [181].

## 4. Conclusions

### 4.1. Regulation Systems Involved in all Cellular Processes

As seen throughout the previous section, TCSs are involved and play a key role in virtually all cellular processes of *S. coelicolor*. This includes morphological development and differentiation (BldM [66,67,68,69,70], WhiI [70,174,175,176,177,178], and RamR [153,154], among others), primary metabolism (PhoR/P [28,103,104,105,106,108,133,134,135,136,137,138,139,140,141,142,143,144,145,146,147,148,149,150,151] and GlnR [48,88,89,90,91,92,93,94,95,96,97,98,99,100,101,102,103,104,105,106,107,108,109], among others), and secondary metabolism (RedZ [156,157,158], AbsA1/A2 [51,52,53,54,55,56,57,58,59,60,61], and EcrA1/A2 [86], among others), as well as in the response to different types of stress such as oxidative stress (SenS/R [160]), osmotic stress (OsaA/B [125,126]), and secretion stress (CssS/R [77]), to give some examples. The importance of TCSs in the physiology of this organism is predictable, especially considering they are one of the main prokaryotic transduction systems.

Although some of the TCSs described are involved in a single cellular process, such as VanS/R in resistance to vancomycin [161] and ChiS/R in the regulation of ChiC chitinase [71,72], most of them are frequently involved in a multitude of biological processes, acting as pleiotropic regulators and essential pillars for their integration and coordination. In this sense, it is important to highlight that the signaling cascades of many of these regulatory systems overlap since they share target genes. This association allows the same process to be differentially regulated based on the various environmental conditions that the organism encounters (e.g., MtrB/A controls the assimilation of nitrogen in media rich in this element, while GlnR does so in limiting conditions). Furthermore, the most necessary cellular processes in different situations can be prioritized (for example in the case of PhoR/P, which can block much of the metabolism and development of the organism in phosphate-limiting situations). It should also be mentioned that most of the cross-regulation events described occur for those TCSs involved in the control of primary metabolism such as PhoP/R, GlnR, MtrB/A, and AfsQ2/Q1 [48,102,103,104,105,106,107,108,109,110,111,112,113,114,115,116,117].

The complex regulatory network formed by these systems and other cellular regulators, as well as their dependence on environmental conditions, makes the study and understanding of TCSs at a more global level significantly difficult.

### 4.2. Molecular Mechanism of Action and its Modulation

Many of the TCSs of this organism have a canonical organization and architecture, as well as a typical molecular mechanism of action. However, some exceptions may be of great interest from a more general point of view, regarding how this type of regulatory system works, and from a more specific point of view, regarding the manipulation and handling of this organism and others that are similar.

In *S. coelicolor*, there are three hybrid HKs (OsaA [125,126], SCO4009, and SCO7327) whose signaling cascades are unclear. In all three cases, they have REC domains that contain a phosphorylatable Asp residue. Moreover, the existence of accessory HPt proteins acting as intermediates in the phosphotransfer process toward the RR are common. However, in principle, there are no proteins of this type in *S. coelicolor*. The reason for this may involve an alternative signaling pathway for these systems, perhaps interacting with other signaling cascades or forming oligomers with certain RRs to modulate their action. Hence, it would be interesting to study these systems in more depth. Additionally, it is noteworthy that in all three cases, the HK lacks a transmembrane helix (so they might be cytosolic). Moreover, similar to OsaA, the associated OsaB RR has a peculiar effector domain (coiled coil) that does not belong to any of the classical families and whose mechanism of action has not yet been fully elucidated [125,126].

Additional HKs of potential interest are those belonging to conservons. These oHKs can interact with other proteins of these operons to give rise to complexes similar to eukaryotic GPCRs. Although several of these proteins (CvnA1, CvnA9, and CvnA10) have been studied [79,80], their biological role and their involvement in processes such as antibiotic production or morphological development, their molecular mechanism of action, and their signaling cascade, which could be of great interest, have not yet been analyzed in depth.

The different TCSs described exhibit a wide variety of mechanisms for regulating the activation state of the RR beyond the canonical phosphorylation of its Asp residue. This is relatively evident in atypical RRs, which lack key residues from the phosphorylation site, justifying additional modulation mechanisms such as the end product that regulates its own signaling pathway (such as RedZ by the RED molecule [157,158]), which can be useful in these signal transduction pathways. However, this is more significant in those RRs that, despite having a canonical phosphorylation site, present alternative mechanisms, as is the case of BldM [66] and GlnR [92]. In the case of BldM, it may be related to its differential oligomerization capacity (formation of homodimers or heterodimers with WhiI), which allows it to act on different sets of genes at different times of the life cycle. In the case of GlnR, post-translational modifications, such as the phosphorylation of Ser and Thr residues, as well as the acetylation of Lys, have been described, which vary depending on the levels of nutrients in the medium and modulate the binding affinity to DNA, that is, the response generated [88]. Regarding this mode of regulation, it would be interesting to see if it is present in other RRs, especially those related to cellular metabolism.

Regarding orphan systems, oHK and oRR that are associated with each other have been described such as OhkA/Orra [42] and PdtaS/R [41], but there are also others that can intervene in the signaling cascades of other TCSs. This is especially relevant in some cases where completely different cellular responses are generated, which is an additional point of regulation; an example of this is OsdK. In its signaling cascade involving OsdR (its associated RR), it acts as a positive regulator of ACT production [129], while in the one involving oRR SCO3818, it acts as a negative regulator of ACT production [43].

Finally, it should be noted that the modulation of the signaling cascades of TCSs is essential for the organism, since a mismatch in these cascades could lead to serious problems in cell function, including cytotoxicity, as has been described, e.g., due to the accumulation of AbrA2 and AbrC3 RRs in their phosphorylated forms [45,49]. Therefore, there are many control mechanisms such as the self-regulation of TCSs AbrA1/A2 [45], AbsA1/A2 [53,55], CseC/B [74], DraK/R [81], PhoR/P [135,146], RedZ [156], and WhiI [178], among others.

### 4.3. Biotechnological Application

One of the main interests in the study of *S. coelicolor* has been to elucidate the regulatory mechanisms of antibiotic production. As previously mentioned, most of the TCSs described in this organism participate in the control of the synthesis of these compounds (Figure 3), either directly through the control of pathway-specific regulators or genes of the biosynthetic cluster, or indirectly. In fact, some of these systems, such as RedZ and AbsA1/A2, are part of some of these biosynthetic clusters.

The role of different TCSs in controlling the production of secondary metabolites of interest is highly conserved in other Streptomycetes. For example, DraK/R modulates the production of ACT and RED in *S. coelicolor*, as well as avermectin and oligomycin in *S. avermitilis* [81]. MtrB/A modulates the production of ACT, RED, and CDA in *S. coelicolor*; chloramphenicol and jadomycin in S. *venezuelae*; avermectin and oligomycin in *S. avermitilis*; and validamycin in *S. hygroscopicus* [116].

It has been widely shown that the biotechnological potential of these systems has two main applications. On the one hand, their manipulation can lead to host strains with an enhanced production capacity of compounds of interest. For example, the removal of the negative regulator AbrA1/A2 of antibiotic production represents a significant improvement in the heterologous production of oviedomycin in *S. coelicolor* [45]. On the other hand, its manipulation or expression in other strains may be key for activating cryptic biosynthetic clusters and for discovering new molecules of interest, as has been shown by the heterologous expression of the AbsA1 HK [61] and the AbrC3 RR in other species [182].

With respect to the use of *Streptomyces* strains in the production of compounds of interest on an industrial level, it is also essential to characterize the TCSs involved in primary metabolism, as well as in morphological development and differentiation, to optimize their growth in industrial fermenters (in fact, some systems such as SCO5282/83 have been described precisely for this purpose [179]), or in other situations, such as agricultural biocontrol agents.

Recent research on TCSs has shown that some are involved in antibiotic resistance such as VanS/R [161], CsecC/B [73], and AbrB1/B2 [46]. This discovery has presented another potential application for this model species, as using the information obtained with their study not only improves the understanding of antibiotic resistance mechanisms in Streptomycetes, but also the highly relevant human pathogens such as *S. aureus* or *E. faecium.*

Finally, another potential biotechnological application for *S. coelicolor* TCSs is to engineer proteins to obtain chimeric HKs (HKs resulting from the fusion of the domains of different HKs). The resulting proteins can therefore be used as biosensors in other organisms [183]. Furthermore, chimeric HKs could generate cellular responses of great biotechnological interest, such as LinR, i.e., the chimeric HK generated from AbrC1 and AbrC2 that confers resistance to lincomycin [50].

### 4.4. Future Considerations

Despite the large number of studies that have been carried out in this field, much remains to be revealed regarding the TCSs discussed in this review, both in terms of the physiological role they play as well as the signaling cascades and mechanisms of action they present. The identification of the activation signals of these systems and the structural characterization of resulting proteins are some of the future tasks that must be undertaken.

The complex regulatory network formed by the TCS in *S. coelicolor* is far from being unraveled. Many of the systems that have already been described still require further examination and the functions of approximately two-thirds of TCSs remain unknown. Nevertheless, the biotechnological potential of these bacteria has been shown to be enormous in recent decades and will continue to be in the future.

## Figures and Tables

**Figure 1 ijms-23-15085-f001:**
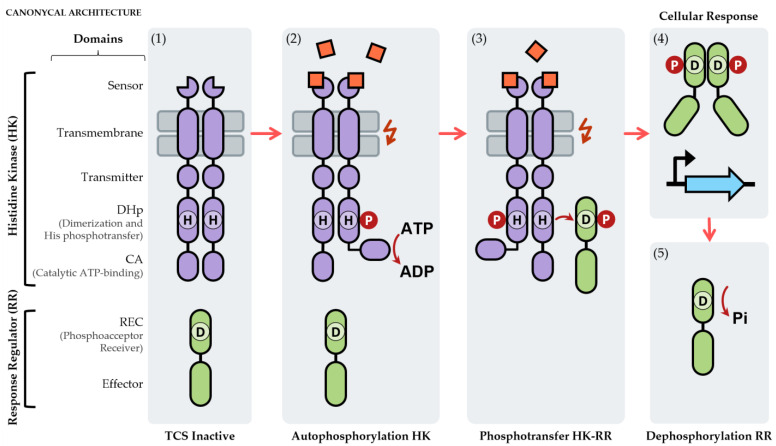
Molecular mechanism of action of a two-component system: (1) if the activation signal is absent, the TCS remains inactive; (2) when the signal appears, the HK autophosphorylates a His residue (H) of the dimerization/phosphotransfer domain (DHp) through the catalytic ATP-binding domain (CA), which requires ATP; (3) the HK transfers the phosphoryl group via its His to an Asp residue (D) of the REC domain of the RR; (4) the RR generally dimerizes and triggers the cellular response, usually through the regulation of gene expression; (5) finally, the signaling cascade inactivates thanks to the dephosphorylation of the RR, usually through its own HK. The different domains of the canonical HK and RR are indicated on the left-side of the figure.

**Figure 2 ijms-23-15085-f002:**
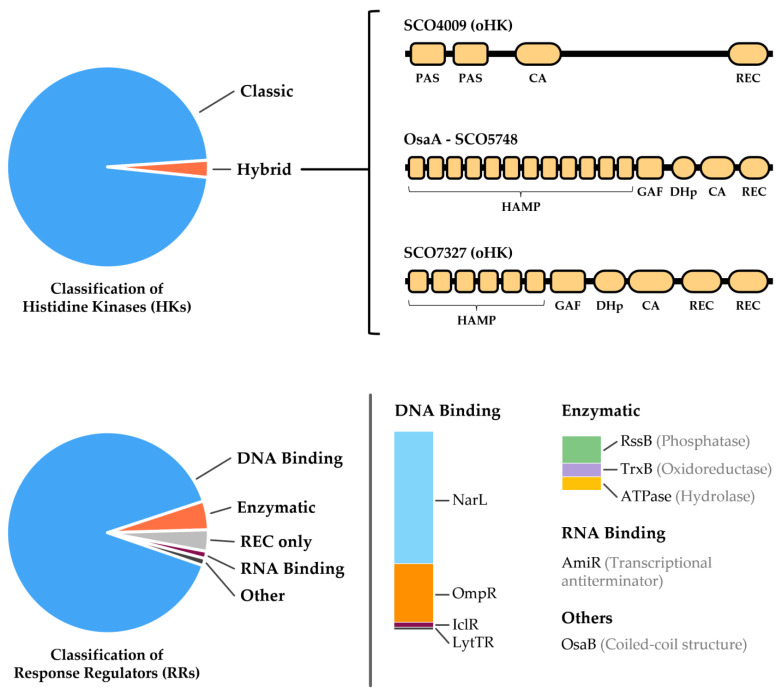
Classification of HKs (top image) and RRs (bottom image) from *S. coelicolor*. In the case of the HKs, the predicted domain architectures of hybrid HKs are shown (based on InterPro). With respect to the RR, the subfamilies of each type of effector domain are indicated. Data were collected from the P2CS and P2RP databases.

**Figure 3 ijms-23-15085-f003:**
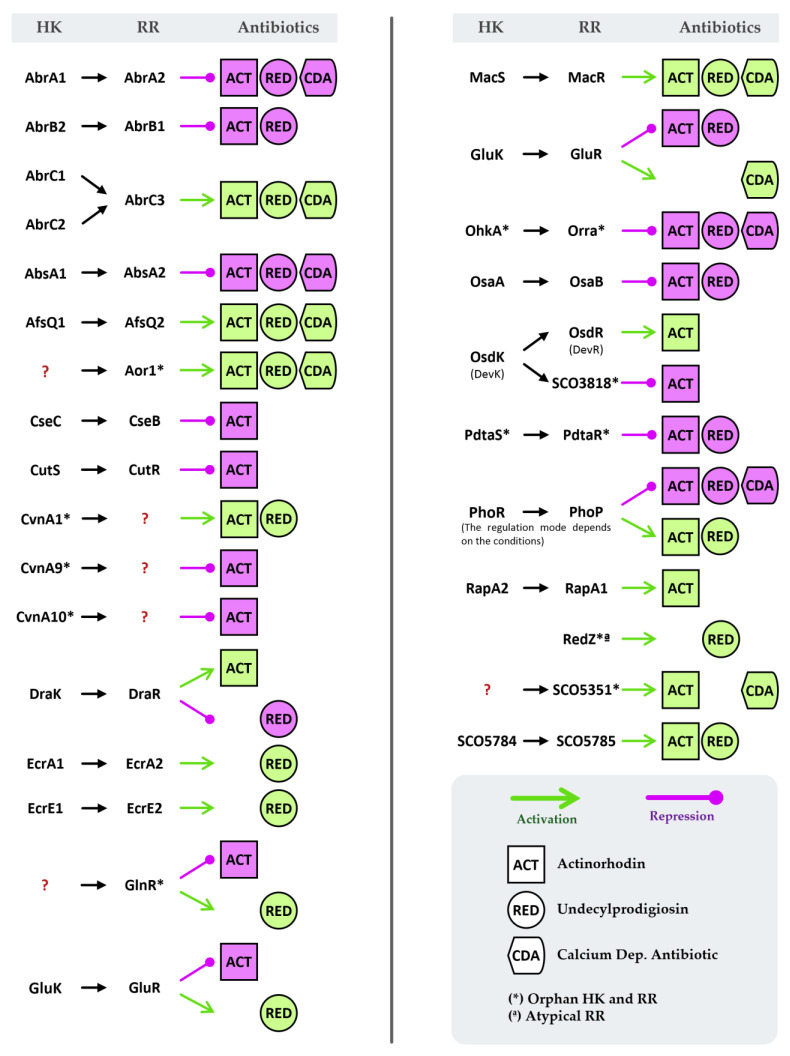
Regulation of antibiotic production in *S. coelicolor* by TCSs.

**Table 1 ijms-23-15085-t001:** TCSs of *S. coelicolor*. All HKs and RRs identified in this bacterium are indicated. They are listed in the SCO number and have been organized based on the pairs or triads they form at the genomic level since it usually indicates a functional association (orphans are indicated individually). The family to which they belong is included; in the case of the HKs, this is based on the architecture of the protein domains, while for the RRs, the type of effector domain is taken in account. The order in which the genes are found at the genomic level is also indicated.

Histidine Kinase (HK)			Response Regulator (RR)			Organization (5′→3′)
Gene	Name	Family	Gene	Name	Family	
*SCO0203*	OsdK/DevS	Classic	*SCO0204*	OsdR/DevR	NarL	RR-HK
*SCO0211*	-	Classic				oHK
*SCO0422*	-	Classic	*SCO0421*	-	NarL	HK-RR
*SCO0551*	-	Classic	*SCO0552*	-	OmpR	RR-HK
*SCO0588*	CvnA11	Classic				oHK
*SCO0599*	-	Classic				oHK
*SCO0676*	-	Classic				oHK
*SCO0767*	-	Classic				oHK
*SCO0871*	-	Classic	*SCO0870*	-	CheY	RR-HK-RR
			*SCO0872*	-	RssB	
*SCO0946*	-	Classic				oHK
*SCO1071*	-	Classic	*SCO1070*	-	NarL	HK-RR
*SCO1137*	-	Classic	*SCO1136*	-	IclR	HK-RR
*SCO1160*	CvnA3	Classic				oHK
*SCO1217*	-	Classic				oHK
			*SCO1220*	-	LytTR	oRR
*SCO1259*	-	Classic	*SCO1260*	-	NarL	HK-RR
*SCO1369*	-	Classic	*SCO1370*	-	NarL	HK-RR
*SCO1402*	CvnA4	Classic				oHK
*SCO1596*	OhkA	Classic				oHK
*SCO1630*	CvnA9	Classic				oHK
			*SCO1654*	-	NarL	oRR
*SCO1744*	AbrA1	Classic	*SCO1745*	AbrA2	NarL	HK-RR
*SCO1802*	-	Classic	*SCO1801*	-	NarL	HK-RR
			*SCO2013*	PdtaR	AmiR	oRR
*SCO2121*	MacS	Classic	*SCO2120*	MacR	NarL	HK-RR
*SCO2142*	-	Classic	*SCO2143*	-	OmpR	HK-RR
			*SCO2152*	-	CheY	oRR
*SCO2166*	AbrB2	Classic	*SCO2165*	AbrB1	NarL	HK-RR
*SCO2215*	-	Classic	*SCO2216*	-	NarL	HK-RR
			*SCO2281*	Aor1	NarL	oRR
*SCO2307*	-	Classic	*SCO2308*	-	NarL	HK-RR
*SCO2359*	-	Classic	*SCO2358*	-	NarL	HK-RR
*SCO2452*	-	Classic				oHK
*SCO2518*	EcrA1	Classic	*SCO2517*	EcrA2	NarL	HK-RR
*SCO2800*	-	Classic	*SCO2801*	-	OmpR	RR-HK
*SCO2879*	CvnA12	Classic				oHK
			*SCO3008*	OrrA	NarL	oRR
*SCO3012*	MtrB	Classic	*SCO3013*	MtrA	OmpR	RR-HK
*SCO3062*	DraK	Classic	*SCO3063*	DraR	OmpR	RR-HK
*SCO3119*	-	Classic				oHK
			*SCO3134*	-	NarL	oRR
			*SCO3144*	-	NarL	oRR
*SCO3225*	AbsA1	Classic	*SCO3226*	AbsA2	NarL	HK-RR
*SCO3284*	-	Classic				oHK
*SCO3359*	CseC	Classic	*SCO3358*	CseB	OmpR	RR-HK
*SCO3390*	SatK	Classic	*SCO3389*	SatR	NarL	HK-RR
*SCO3589*	VanS	Classic	*SCO3590*	VanR	OmpR	RR-HK
*SCO3639*	-	Classic	*SCO3638*	-	NarL	HK-RR
*SCO3641*	-	Classic	*SCO3640*	-	NarL	HK-RR
*SCO3654*	-	Classic	*SCO3653*	-	NarL	HK-RR
*SCO3740*	-	Classic	*SCO3741*	-	OmpR	RR-HK
*SCO3750*	-	Classic				oHK
*SCO3757*	-	Classic	*SCO3756*	-	NarL	HK-RR
*SCO3796*	-	Classic				oHK
			*SCO3818*	-	NarL	oRR
*SCO3948*	-	Classic				oHK
*SCO4009*	-	Hybrid				oHK
*SCO4021*	-	Classic	*SCO4020*	-	OmpR	RR-HK
*SCO4073*	RagK	Classic	*SCO4072*	RagR	NarL	HK-RR
*SCO4120*	-	Classic				oHK
*SCO4124*	-	Classic	*SCO4123*	-	NarL	HK-RR
*SCO4155*	CssS	Classic	*SCO4156*	CssR	OmpR	RR-HK
			*SCO4159*	GlnR	OmpR	oRR
			*SCO4201*	-	RssB	oRR
*SCO4229*	PhoR	Classic	*SCO4230*	PhoP	OmpR	HK-RR
*SCO4275*	SenS	Classic	*SCO4276*	SenR	NarL	HK-RR
*SCO4362*	-	Classic	*SCO4363*	-	NarL	HK-RR
*SCO4597*	AbrC2	Classic	*SCO4596*	AbrC3	NarL	HK-HK-RR
*SCO4598*	AbrC1	Classic				
*SCO4667*	SitK	Classic	*SCO4668*	SitR	NarL	HK-RR
			*SCO4768*	BldM	NarL	oRR
*SCO4791*	-	Classic	*SCO4792*	-	NarL	HK-RR
*SCO4906*	AfsQ2	Classic	*SCO4907*	AfsQ1	OmpR	RR-HK
			*SCO5006*	-	ATPase	oRR
*SCO5040*	-	Classic				oHK
*SCO5104*	-	Classic				oHK
*SCO5131*	-	Classic	*SCO5132*	-	NarL	HK-RR
*SCO5239*	PdtaS	Classic				oHK
*SCO5282*	-	Classic	*SCO5283*	-	OmpR	RR-HK
*SCO5289*	CvnA5	Classic				oHK
*SCO5304*	-	Classic				oHK
			*SCO5351*	-	CheY	oRR
*SCO5378*	ChiS	Classic	*SCO5377*	ChiR	NarL	HK-RR
*SCO5404*	RapA2	Classic	*SCO5403*	RapA1	OmpR	RR-HK
*SCO5435*	-	Classic	*SCO5434*	-	IclR	HK-RR
*SCO5454*	-	Classic	*SCO5455*	-	NarL	HK-RR
*SCO5460*	-	Classic				oHK
*SCO5540*	CvnA2	Classic				oHK
*SCO5544*	CvnA1	Classic				oHK
*SCO5683*	-	Classic	*SCO5684*	-	NarL	HK-RR
*SCO5748*	OsaA	Hybrid	*SCO5749*	OsaB	OsaB	HK-RR
*SCO5779*	GluK	Classic	*SCO5778*	GluR	OmpR	RR-HK
*SCO5784*	-	Classic	*SCO5785*	-	NarL	HK-RR
*SCO5824*	-	Classic	*SCO5825*	-	NarL	HK-RR
*SCO5829*	-	Classic	*SCO5828*	-	NarL	HK-RR
*SCO5863*	CutS	Classic	*SCO5862*	CutR	OmpR	RR-HK
*SCO5871*	KdepD	Classic	*SCO5872*	KdpE	OmpR	HK-RR
			*SCO5881*	RedZ	NarL	oRR
			*SCO6029*	WhiI	NarL	oRR
*SCO6069*	CvnA6	Classic				oHK
*SCO6139*	-	Classic	*SCO6140*	-	NarL	HK-RR
*SCO6163*	-	Classic	*SCO6162*	-	NarL	HK-RR
*SCO6253*	-	Classic	*SCO6254*	-	NarL	HK-RR
*SCO6268*	-	Classic				oHK
*SCO6353*	-	Classic	*SCO6354*	-	OmpR	RR-HK
*SCO6362*	-	Classic	*SCO6363*	-	NarL	HK-RR
			*SCO6364*	-	OmpR	oRR
*SCO6369*	-	Classic				oHK
*SCO6421*	EcrE1	Classic	*SCO6422*	EcrE2	NarL	HK-RR
*SCO6424*	-	Classic				oHK
*SCO6484*	-	Classic				oHK
*SCO6668*	-	Classic	*SCO6667*	-	NarL	HK-RR
			*SCO6685*	RamR	NarL	oRR
*SCO6794*	CvnA7	Classic				oHK
*SCO6943*	CvnA8	Classic				oHK
*SCO7009*	-	Classic				oHK
*SCO7076*	-	Classic	*SCO7075*	-	OmpR	RR-HK
*SCO7089*	-	Classic	*SCO7088*	-	NarL	HK-RR
*SCO7220*	-	Classic				oHK
*SCO7231*	-	Classic	*SCO7230*	-	OmpR	RR-HK
*SCO7297*	-	Classic	*SCO7298*	-	TrxB	RR-HK
*SCO7327*	-	Hybrid				oHK
*SCO7354*	-	Classic				oHK
*SCO7422*	CvnA10	Classic				oHK
*SCO7463*	CvnA13	Classic				oHK
*SCO7534*	-	Classic	*SCO7533*	-	OmpR	RR-HK
*SCO7649*	-	Classic	*SCO7648*	-	NarL	HK-RR
*SCO7711*	-	Classic	*SCO7712*	-	NarL	HK-RR

## Data Availability

Not applicable.

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
