# Peer review of "Two-Component Systems of Streptomyces coelicolor: An Intricate Network to Be Unraveled"

_ijms, 2022, doi:10.3390/ijms232315085_

Round 1

Reviewer 1 Report

This manuscript provides an overview of the two-component system of Streptomyces coelicolor. The presentation of the two-component system and its regulatory mechanisms is very comprehensive and detailed throughout. The figure in the manuscript is also concise and clear.

My suggestions for the manuscript are as follows:

 1.    The table 1 should be annotated for ease understanding.

2.    The conclusion part should give a more comprehensive overview of the whole manuscript.

3.    Please check the writing rules carefully. For example, Journal name for ref. 81 and 82.

4.    The order of 3.1 to 3.36 could not find the logic. So it is suggested to order the importance of the two-component system.

5.     Many regulators have been described. However, they were scattered. There is only a list of literature in the whole manuscript, but no in-depth discussion and summary of the results.

6. In your opinion, what are their potential biotechnological applications?
7. Usually, the same genus name is usually abbreviated when it does not appear for the first time. For example, Staphylococcus aureus can be abbreviated as S. aureus.

Author Response

This manuscript provides an overview of the two-component system of Streptomyces coelicolor. The presentation of the two-component system and its regulatory mechanisms is very comprehensive and detailed throughout. The figure in the manuscript is also concise and clear.

My suggestions for the manuscript are as follows:

  1. The table 1 should be annotated for ease understanding.

We have included more annotations in the table title to make it easier to understand.

  1. The conclusion part should give a more comprehensive overview of the whole manuscript.

We have expanded the part of the conclusion for a more in-depth discussion of the regulatory systems covered in this review.

  1. Please check the writing rules carefully. For example, Journal name for ref. 81 and 82.

We have revised the bibliography to correct formatting errors.

  1. The order of 3.1 to 3.36 could not find the logic. So it is suggested to order the importance of the two-component system.

Due to the great diversity of systems, we have decided to order them in alphabetical order of the assigned name (and at the end of all, those systems that have not been assigned a name beyond the SCO). A sentence indicating the order chosen has been added

Ordering them by importance based on, for example, the number of publications about them, although it may be clearer in some cases (such as PhoP/R, GlnR or VanS/R) it might be more problematic in many others that present a similar presence in literature. So, in the end we consider that it could be more confusing.

We also considered sorting them by the cellular processes they are involved in or whether they are orphans or not, but it was also more challenging due to the large number of systems.

That is why we consider that the clearest way to present them was the alphabetical order.

  1. Many regulators have been described. However, they were scattered. There is only a list of literature in the whole manuscript, but no in-depth discussion and summary of the results.

As we commented in the second point, we have expanded the conclusions.

  1. In your opinion, what are their potential biotechnological applications?

It is included in the new conclusions.

  1. Usually, the same genus name is usually abbreviated when it does not appear for the first time. For example, Staphylococcus aureus can be abbreviated as S. aureus.

We have corrected the errors in the abbreviations.

Reviewer 2 Report

As the title of the manuscript by Sánchez de la Nieta et al., indicates “Two-Component Systems of Streptomyces coelicolor: An Intricate Network to be Unravelled”, it is focused on an exhaustive description of the TCSs present in this bacterium. One of the main interests of this microorganism is that it is “the main model species, as it is one of the first to be morphologically and genetically characterized” from the Streptomyces genus, which “constitutes an authentic biotech gold mine thanks to their ability to produce a myriad of compounds and enzymes of great interest at the clinical, agricultural, and industrial levels”. I agree with the authors in the relevance (scientific, technological, …) of these bacteria.

            As it is, I find two distinct parts in this manuscript: sections 1 and 2, and section 3, respectively. Sections 1 and 2 are really readable, informative, with updated references and, I am sure, highly interesting for the general readers of this journal. However, section three is different in that it is prepared for scientists within the field of TCSs. I understand that this is a review and therefore it is not uncommon to see such a collection of specific information; but in this particular case, this large section is a pseudo-encyclopedic text. As such, I feel that it is a simple summary of data, without elaboration. This lack of elaboration is the reason why the “Conclusions” section is poorly informative. My recommendation to the Editor is "reject".

Author Response

As the title of the manuscript by Sánchez de la Nieta et al., indicates “Two-Component Systems of Streptomyces coelicolor: An Intricate Network to be Unravelled”, it is focused on an exhaustive description of the TCSs present in this bacterium. One of the main interests of this microorganism is that it is “the main model species, as it is one of the first to be morphologically and genetically characterized” from the Streptomyces genus, which “constitutes an authentic biotech gold mine thanks to their ability to produce a myriad of compounds and enzymes of great interest at the clinical, agricultural, and industrial levels”. I agree with the authors in the relevance (scientific, technological, …) of these bacteria.

As it is, I find two distinct parts in this manuscript: sections 1 and 2, and section 3, respectively. Sections 1 and 2 are really readable, informative, with updated references and, I am sure, highly interesting for the general readers of this journal. However, section three is different in that it is prepared for scientists within the field of TCSs. I understand that this is a review and therefore it is not uncommon to see such a collection of specific information; but in this particular case, this large section is a pseudo-encyclopedic text. As such, I feel that it is a simple summary of data, without elaboration. This lack of elaboration is the reason why the “Conclusions” section is poorly informative. My recommendation to the Editor is "reject".

We believe that this review can be of great interest, considering that the last ones carried out exhaustively on this topic are several years old and are extremely outdated.

Regarding section 3, we have obviously included the specific information necessary for this article to be a review on TCS of S.coelicolor, and not a simple general review of two components systems of which there are already other very good reviews. The fact that it is such a long section derives from the fact that there are many TCS described in this strain (36 systems, with almost 200 references). Therefore, although they are briefly commented, it takes up a lot of space and the more important works in each system are referred. In that sense, we consider quite debatable the lack of elaboration that you indicate.

In the current revision of the manuscript we have expanded the conclusions section, the biotechnological aplications and made some other small changes according to the comments of both reviewers. Attending to these improvements and to what you indicate in the revision, “that the topic is very relevant”, and that part of the manuscript is, in your own words, "really readable, informative, with updated references and highly interesting for the general readers of this journal", we hope that you reconsider your previous decision in order to accept it in its actual form.

Round 2

Reviewer 2 Report

In this second version of the manuscript, the authors have done a significant effort to integrate a great pool of information presented in section 3 and have prepared a final section (Conclusions) that changes the complete work from a mainly record of scattered information (of course, highly interesting for experts in the field of TCSs) to a highly informative review for the more general reader interested in the topic.

Just a minor point: as it is prepared, I would change the title of section 4 to indicate that it is presented an integrative view of the TCSs from S. coelicolor.